# Microcirculation-Promoting Effect of Escin on Cutaneous Tissue via Gsk3β Down-Regulation

**DOI:** 10.3390/cimb47100840

**Published:** 2025-10-14

**Authors:** Jaeyoon Kim, Jang Ho Joo, Heena Rim, Sung Hyun Kim, Jae young Shin, Seung-Hyun Jun, Nae-Gyu Kang

**Affiliations:** Household & Health Care (LG H&H) R&D Center, 70, Magokjoongang 10-ro, Gangseo-gu, Seoul 07795, Republic of Korea; kjy5281@lghnh.com (J.K.); janghojoo@lghnh.com (J.H.J.); hina751@lghnh.com (H.R.); ness9510@lghnh.com (S.H.K.); sjy2811@lghnh.com (J.y.S.); junsh@lghnh.com (S.-H.J.)

**Keywords:** escin, Wnt/β-catenin pathway, microcirculation, blood vessel remodeling

## Abstract

Microcirculation in cutaneous tissue is essential to balance oxygen delivery and maintain the health of the skin. Senescence contributes to microcirculatory dysfunction through mechanisms involving chronic inflammation, structural remodeling of microvessels, and disturbances in hemodynamics. In this study we investigated the promoting effect of escin on blood flow through topical application. To elucidate the molecular mechanisms of escin, kinase phosphorylation changes in human umbilical vein endothelial cells (HUVECs) were examined. Escin stimulates the Wnt/β-Catenin and c-Jun N-terminal kinase (JNK) signaling pathway in cultured HUVECs. To clarify the target of escin in the Wnt/β-Catenin signaling pathway, gene expression in response to escin treatment was evaluated, and escin-mediated signaling activation was accompanied by glycogen synthase kinase-3 beta (Gsk3β), according to inhibitor studies performed with IWR1 (tankyrase inhibitor). In addition, the expression level of the Gsk3β were down-regulated by escin treatment in cultured HUEVCs. Escin also enhanced vascular remodeling, and, when applied topically, led to a sustained increase in cutaneous blood flow. Escin-mediated Wnt signaling activation could enhance blood vessel networks via Gsk3β down-regulation. In conclusion, our data demonstrate that escin promotes angiogenic behavior and enhances adenosine-induced perfusion in humans, thereby supporting its potential role in modulating cutaneous microcirculation.

## 1. Introduction

Skin, as the body’s outermost organ, serves a critical role in protecting the host from various environmental stressors, including toxins, UV radiation, and pathogenic microorganisms [1]. Within this system, arterioles play a crucial role in the regulation of blood flow [2]. The microcirculation of cutaneous tissue encompasses vessels that are imperceptible to the naked eye, specifically the smallest resistance arteries (less than 150 μm in diameter), arterioles, capillaries, and venules [3]. The intricate mechanisms governing skin blood-flow regulation involve several processes, including the myogenic response of arterioles, endothelium-dependent flow-induced vasodilation mediated by factors such as nitric oxide, prostacyclin, and endothelium-dependent hyperpolarizing factors, alongside metabolic and neurovascular interactions [4]. Notably, sympathetic activation triggers vasoconstriction through the secretion of norepinephrine, while a sympathetic cholinergic system facilitates vasodilation [5]. Additionally, other neuronal types release vasodilatory substances such as calcitonin gene-related peptide (CGRP) or substance P [6]. The variations observed in skin blood flow stem from complex interactions among these regulatory mechanisms [7].

Vascular dysfunction manifests as a systemic disease process characterized by impaired endothelium-dependent vasodilation, increased vasoconstriction, and structural alterations within microvessels, occurring simultaneously across various vascular beds [8]. The remodeling of vascular architecture and the corresponding reduction in endothelial-derived vasodilators, such as nitric oxide (NO), may represent some of the earliest indicators of cardiovascular disease [9]. Recent studies have identified the skin’s vascular system as a valuable and accessible model for examining the mechanisms that underlie both microcirculatory function and dysfunction [10]. The skin has been utilized for investigating vascular mechanisms associated with several pathological conditions, including hypercholesterolemia, hypertension, Type II diabetes, peripheral vascular disease, heart failure, systemic sclerosis, alopecia, dermatological disorders, and primary aging [11].

Recent studies highlight the critical function of the Wnt signaling pathway in vascular morphogenesis during embryonic development and recovery of adult tissue within specific organs [12]. Wnt signaling is essential for various developmental processes, including cell fate determination, cell proliferation and survival, and the overall formation of organs [13]. The Wnt factors utilize multiple receptors and distinct signaling pathways. The most extensively studied Wnt signaling cascade involves the transcriptional activity of β-catenin [14]. When Wnt ligand binds to the Frizzled/Lrp receptor complex, it disrupts the degradation complex, leading to the stabilization of β-catenin in the cytoplasm. This stabilization facilitates the translocation of β-catenin into the nucleus, where it engages with T-cell factor (TCF)/(Lef) transcription factors to initiate gene transcription [13]. Rescuing the Wnt/β-catenin signaling pathway would be a promising strategy for treating and/or preventing cutaneous diseases [12,15]. In consequence, diverse synthetic and natural Wnt activators, such as retinoic acid and berberine, have been investigated [16,17].

Escin is a combination of triterpene saponins extracted from the seeds of *Aesculus hippocastanum* [18]. It has been extensively utilized in the treatment of various ailments such as cough, diarrhea, dysmenorrhea, and bladder conditions in ethnopharmacological studies [19]. Furthermore, preclinical investigations have shown that escin reduces extravasation and immune cell infiltration [20], lowers histamine levels [21], inhibits cytokine secretion [22], and maintains the functionality of endothelial cells (EC) in tissue [23].

Recent research and evidence has indicated that escin’s antivaricose properties can be linked to its anti-inflammatory and anti-edematous effects on ECs [24]. Clinical applications of escin’s anti-inflammatory effects in diverse tissues have also been reported [25]. For instance, escin provides protective benefits to the blood–brain barrier and neurological health by mitigating systemic inflammation through the NF-κB signaling pathways [26]. Escin has been investigated as a potential therapeutic agent for vascular disorders and has been demonstrated to alleviate chemotherapy-induced phlebitis through inhibition of the GBP5/NLRP3 axis. [27]. In previous reports, it has found that escin is a natural Wnt agonist and stimulates Wnt/β-Catenin signaling by facilitating the proteasomal degradation of Gsk3β [28]. Nevertheless, the mechanism behind its anti-edematous and anti-inflammatory effects remains unclear.

In this study, the molecular mechanisms underlying the anti-edematous activity of escin and the effect of blood flow enhancement on topical administration were investigated. The kinase phosphorylation changes in cultured human umbilical vein endothelial cells (HUVECs) were examined. To clarify the effect of escin on the Wnt/β-Catenin signaling pathway, gene expression in response to escin treatment was evaluated, and an inhibitor study with Wnt signaling antagonists, DKK1, IWR1, and Wnt-C59, was performed. In addition, blood vessel formation assays were performed. Topical administration of escin was also investigated, and the enhancement of blood flow in cutaneous tissue was evaluated. This study aimed to elucidate the mechanisms by which escin modulates vascular remodeling and blood flow, providing insight into its potential role in cutaneous microcirculation.

## 2. Materials and Methods

### 2.1. Human Umbilical Vein Endothelial Cell Culture

Human umbilical vein endothelial cells (HUVECs) were obtained from PromoCell (Heidelberg, Germany). HUVECs were cultured in EGM-2 medium (Lonza, Basel, Switzerland). Cells were maintained in a humidified incubator at 37 °C, 5% CO_2_. Human recombinant DKK1 were purchased from R&D Systems (Minneapolis, MN, USA). IWR1 and Wnt-C59 were purchased from Tocris Bioscience (Bristol, UK).

### 2.2. Cell Viability Assay for Cultured HUVECs

The cell viability of human umbilical vein endothelial cells (HUVECs) following escin treatment was assessed using the Cell Counting Kit-8 (CCK-8; Dojindo, Rockville, MA, USA) according to the manufacturer’s instructions. Cellular energy metabolism was evaluated by quantifying NAD(P)H after a 4 h exposure to escin (Sigma-Aldrich, St. Louis, MO, USA). Vehicle-treated cells served as untreated controls, and Wnt3a (100 ng/mL) was used as a positive control [29]. Absorbance at 450 nm was recorded with an H1 microplate reader (BioTek, Santa Clara, VT, USA).

### 2.3. Quantitative Real-Time PCR

HUEVCs were exposed to escin at 4, 8, or 16 μM for 4 h; untreated cells served as controls. For inhibitor experiments, DKK1 (2.5–40 μg/mL), IWR-1 (31–500 nM), and Wnt-C59 (0.01–100 nM) were co-administered with escin and Wnt3a (100 ng/mL) as indicated. DKK1 was dissolved in PBS, whereas IWR-1 and Wnt-C59 were dissolved in DMSO; the final DMSO concentration did not exceed 0.5 ppm and had no detectable effects on HUVEC responses.

Total RNA was isolated using the RNeasy kit (Qiagen Inc, Hilden, Germany). First-strand cDNA was synthesized with a cDNA synthesis kit (Philkorea, Seoul, Republic of Korea) on a ThermoCycler (R&D Systems, Minneapolis, MN, USA), following the manufacturers’ instructions. Quantitative real-time PCR was performed using TaqMan chemistry (TaqMan One-Step RT-PCR Master Mix; Life Technologies, Carlsbad, CA, USA) on an ABI 7500 Real-Time PCR System. The following TaqMan assays were used: GAPDH (4352934E), LEF1 (Hs01547250_m1), MYC (Hs00153408_m1), FZD5 (Hs00258278_s1), BMP2 (Hs00154192_m1), and SRD5A2 (Hs00936406_m1). Data were analyzed using instrument software.

### 2.4. TCF/LEF Reporter Assay

WRHEK293A reporter cells (Amsbio, Abingdon, UK) were seeded into black 96-well plates and allowed to adhere for 24 h. Cells were then treated with Wnt pathway activators (Wnt3a and escin) and incubated for an additional 24 h. Lysis was performed by adding 50 μL of 1× Passive Lysis Buffer (Promega, Madison, WI, USA) per well and shaking for 10 min. GFP (internal control for cell number/viability) was quantified by fluorescence (Ex 488 nm/Em 510 nm) on an H1 microplate reader (BioTek, Winooski, VT, USA). Subsequently, 50 μL of luciferase substrate (Promega) was added to each well and luminescence was recorded on the same instrument. Firefly luciferase signals (TCF/LEF reporter activity) were normalized to GFP fluorescence

### 2.5. Phosphorylated Protein Dot Blot Analysis for Human Kinase Signaling Transduction

Phospho-protein profiling was performed using human MAPK and human receptor tyrosine kinase (RTK) phosphorylation antibody arrays (Abcam, Cambridge, UK) per the manufacturer’s instructions, with detection by a pan-anti-phosphotyrosine antibody. In total, 71 RTKs and 17 MAPK pathway components were interrogated. HUVECs were treated with escin (8 or 16 μM) for 4 h and harvested for array analysis; vehicle-treated cells served as untreated controls, and Wnt3a (100 ng/mL) was used as a positive control [29]. For confirmatory immunoblotting, standard procedures were followed, and blots were imaged under identical acquisition settings on an iBright FL1000 system (Thermo Fisher Scientific/Invitrogen, Waltham, MA, USA).

### 2.6. Western Blotting Analysis

Cultured HUVECs (1 × 10^6^ cells/dish) were seeded and allowed to adhere for 24 h. Cells were then treated with escin at 2, 4, 8, or 16 μM for 4 h, after which they were lysed, and total cellular protein was prepared. Equal protein amounts (50 μg) were analyzed by immunoblotting with the following antibodies: GSK-3β (clone 27C10; 1:1000, Cell Signaling Technology, Danvers, MA, USA), phospho-GSK-3β (Ser9; 1:1000, Cell Signaling Technology), and GAPDH (1:2000, Santa Cruz Biotechnology, Santa Cruz, CA, USA). Vehicle-treated cells served as untreated controls, and Wnt3a (100 ng/mL) was used as a positive control [29]. Standard immunoblot procedures were performed according to the manufacturers’ instructions, and blots were imaged under identical acquisition settings on an iBright FL1000 system (Invitrogen, Waltham, MA, USA).

### 2.7. Blood Vessel Formation Assay

Cultured HUVECs (1 × 10^5^ cells/well) were seeded in a µ-Slide 15 Well (ibidi GmbH, Munich, Germany) and cultured with either escin or wnt3a, as a positive control, for 24 h. After the 24h culture, each sample was prepared with Calcein staining (Sigma Aldrich, St. Louis, MO, USA), following the manufacturer’s instructions. Images, with time intervals, were obtained using EVOS^TM^ FL Auto2 Imaging System (Thermofisher Scientific, Waltham, MA, USA). Blood vessel formation rates were calculated using the image segmentation algorithm on the Image J plugin (Version 1.53v), angiogenesis analyzer [30].

### 2.8. Evaluation of Skin Blood Flow In Vivo

This study adhered to the tenets of the Declaration of Helsinki. The study was reviewed and approved by the Institutional Review Board (IRB) of LG Household & Health Care, Ltd. in accordance with the Bioethics and Safety Act (IRB No. LGHH-20241121-AA-03-01; approval date: 2024-11-22). For regulatory context, under the Ministry of Food and Drug Safety (MFDS) Regulations on Safety of Drugs, etc., Article 24 (“Approval of clinical trial plans, etc.”) Paragraph 2 (“Ministry of Food and Drug Safety medicines clinical trials plan approval exception Target”), this study qualifies as approval-exempt and is therefore not subject to clinical-trial plan approval or registration. Written informed consent was obtained from all participants before any study-related procedures. In total, 31 Korean volunteers participated in this study (Age: 28~42, Gender: 19 Male/12 Female). All participants without significant scalp conditions (pruritus, erythema, and alopecia) were enrolled. Skin blood flow was evaluated on the temple. Blood flow in cutaneous tissues was recorded in triplicate using a Periflux system 5000 (Perimed, Las Vegas, NV, USA) [31]. Measurements were performed after blow-drying and resting for 30 min in an air-controlled room.

All volunteers were randomized into four groups: the first group (*n* = 8) used a vehicle, the second group (*n* = 8) used adenosine (0.75%), the third group (*n* = 7) used adenosine (0.75%) with Escin (0.2%), and the fourth group (*n* = 8) used adenosine (0.75%) with Escin (0.5%), using a computer-generated random number system to maintain allocation concealment. The investigators and participants were blinded to the group allocation. The baseline level of all groups was assumed to be statistically equal (*p* = 0.4850, Average: 53.0 ± 16.3 Arbitrary Units), and there was no significant difference between the means of any group. (Appendix A) The participants were supplied with identical bottles containing similar textures, colors, and smells to maintain blinding. The agents (escin and adenosine complex) were applied through daily topical administration (1 mL/day) to the frontal and vertex regions of the scalp. The total investigation period was one month. After one month of administration, a blood flow measurement on the temple was performed. After product treatment, blood flow was measured in a single site, 5 times continuously (baseline, 15, 30, 45, and 60 min). The study followed the CONSTORT statement for reporting randomized clinical trials.

### 2.9. Statistical Analysis

Statistical analyses were performed using Prism 10.0.2 (GraphPad Software, Boston, MA, USA). Data are presented as average ± standard deviation. The data were evaluated using the Shapiro–Wilk test for normal distribution and similar variance between groups. Statistical significance (* *p* < 0.05, ** *p* < 0.01, and *** *p* < 0.001) was evaluated using a two-tailed unpaired Student’s *t*-test for comparisons between two groups and a one-way analysis of variance (ANOVA) with relevant post hoc tests for multiple comparisons. All in vitro experimental data are presented as the average ± standard deviation of at least three independent experiments. For the in vivo study, changes in blood flow within each group were analyzed using one-way repeated measures ANOVA (RM ANOVA). Intergroup comparisons at each time point (15, 30, 45, and 60 min) were performed using a two-tailed unpaired Student’s *t*-test (# *p* < 0.05, ## *p* < 0.01, and ### *p* < 0.001).

## 3. Results

### 3.1. Escin Stimulated Wnt/β-Catenin Signaling

The Wnt/β-catenin pathway is one of the critical signaling pathways in the tissue regeneration process [13]. The possible involvement of escin on the Wnt/β-catenin pathway was investigated in a stable TCF/LEF reporter cell line. The treatment of escin increased luciferase activity in a concentration-dependent manner, indicating that escin acted as an agonist for the TCF/LEF signaling pathway. Wnt3a, a ligand for canonical Wnt/β-Catenin signaling, also increased the reporter activity (Figure 1b).

In addition, to elucidate the effect on cellular energy production and cell survival, cell viability, measured through NAD(P)H generation, was also investigated. The cellular energy production was significantly increased by escin treatment for 4 h (Figure 1c) in cultured HUVEC.

### 3.2. Wnt/β-Catenin Signaling Stimulation by Escin Treatment in Cultured HUVECs

The kinome analysis in HUVECs was further investigated with escin treatment. A total of 18 kinase phosphorylation were significantly up-regulated or down-regulated (more than 1.5-fold or less than 0.7-fold, *p* < 0.001). The kinases, ACK1, AKT, CREB, and HCK, were significantly increased by escin, but Gsk3β was decreased (Figure 2a).

These kinases were selected for further investigation, namely, protein clustering analysis [32] and pathway estimation, following Gene Ontology (GO) and Kyoto Encyclopedia of Genes and Genomes (KEGG) analyses (Figure 2b,c). Escin-mediated signaling activation was related to cellular metabolism and proliferation (Figure 2c). Among these, the PI3K-AKT and longevity regulating pathways were predicted to be stimulated by Escin. In addition, escin stimulated organ development processes, such as the Wnt and mTOR signaling pathways (Figure 2c).

Furthermore, based on in silico modeling, interactions between proteins, such as c-Jun, CTNNB1, and RICTOR, were predicted (Table 1 and Appendix A) [32,33]. In addition, using reverse screening of escin to target proteins through machine learning, c-Jun and p38 were predicted to be potential candidates to show bioactivity (Appendix A) [34,35]. In Figure 3, the phosphorylation of p38 was significantly activated by escin treatment. Therefore, c-Jun activation was also needed to verify the efficacy of escin, as was p38 activation.

### 3.3. Gsk3β Down-Regulation by Escin Treatment in Cultured HUVECs

Following in silico modeling analysis in HUVECs, Gsk3β and c-Jun N-terminal kinases (JNK) were estimated to be one of the proteins most highly correlated with escin treatment. The total expression level of Gsk3β was down-regulated in a dose-dependent manner (Figure 3a). Although phosphorylation of GSK3β at Ser9 was reduced, no significant changes were observed when compared with total GSK3β levels (Appendix A). These findings indicate that the ratio of active to inactive GSK3β remained unchanged, while the overall amount of GSK3β was significantly decreased (Figure 3b). The down-regulation of Gsk3β is one of the well-known targets of Wnt/β-catenin signaling pathway activation [15]. The activation of JNK and its related kinases with escin treatment in vitro was further investigated. Escin treatment produced increased JNK (T183) levels at a concentration of 16 μM (Figure 3c). In addition, the phosphorylation of kinase proteins (MKK3/6 and Gsk3β), which contribute to JNK signaling pathway activation [36], was evaluated (Figure 3d). Phosphorylation of MKK3 (S189) and MKK6 (S207) was significantly increased by escin treatment (Figure 3d,e).

To clarify how escin stimulates the c-Jun and Wnt/β-catenin signaling pathways, we further investigated the changes in mRNA expression in HUVEC treated with escin. The genes BMP2, Lef1, FZD5 and Myc, which have been reported as genes activated by Wnt/β-catenin signaling [37] were significantly increased by escin (Figure 3c).

### 3.4. Escin-Mediated-Wnt/β-Catenin Signaling Was Inhibited by Gsk3β Antagonist IWR1

To elucidate the target of escin in Wnt/β-catenin signaling, an inhibitor study with Wnt signaling antagonists DKK1, IWR1, and Wnt-C59, was performed. It was revealed that escin-mediated activation was abrogated by the treatment of IWR1, a tankyrase inhibitor [38], comparable with that of Wnt3a. On the contrary, DKK1, Lrp5/6 inhibitor [39], and Wnt-C59, PORCN inhibitor [40], did not affect to escin-mediated Wnt signaling activation (Figure 4). Therefore, escin contributed to Gsk3β activity rather than Wnt ligand binding or its production. Taken together, our data suggest that escin stimulate the canonical Wnt/β-catenin signaling pathway and the adenomatous polyposis coli (APC)/Axin/Gsk3β destruction complex is disrupted via Gsk3β inactivation (Figure 4).

### 3.5. Escin Enhanced Vessel Remodeling and Blood Flow In Vivo

To evaluate efficacy of the escin on microcirculation, tube-formation assay and blood flow enhancement in cutaneous tissue were investigated. AP-1 family transcription factors, such as c-Jun, c-Fos, and ATF, are involved in cell proliferation and transformation including blood vessel [41]. Furthermore, Wnt/β-catenin signaling pathway also play pivotal role in angiogenesis and blood vessel remodeling [12]. According to vessel forming assay, escin enhanced blood vessel formation. The number of junction and total segments length were significantly increased by escin treatment (Figure 5a,b) [30].

In addition, cutaneous blood flow enhancement with topical administration of escin was also examined. Topical administration of adenosine evoked elevation of cutaneous blood flow as following previous reports [42]. The vasodilation observed was primarily mediated by adenosine, rather than escin itself, which is consistent with the clinical findings. Acute application of escin did not induce vasodilation, and no short-term changes in blood flow were detected in vivo. However, the long-term application of the escin were enhanced increase in blood flow rate and elongated responses in does dependent manner (Figure 5c,d). Therefore, it is suggested that the mechanisms of escin, activation of Wnt/β-catenin signaling pathway, could contribute to enhance microcirculation of cutaneous tissue.

## 4. Discussion

Aging (senescence) represents an inescapable biological phenomenon that manifests visibly across all cutaneous tissues, notably affecting hair follicles [43]. Skin and hair follicles are the representative organs that could be regenerated after external damage [44]. The restoration of tissue integrity and homeostasis is essential for the survival of organisms [45]. The process of organ recovery and regeneration necessitates a substantial increase in energy demand [46]. Furthermore, effective cellular coordination is crucial for tissue repair. During the stage, complex networks of multiple cytokines and mediators facilitate cellular communication among tissue components, such as keratinocyte and blood vessels [47]. These mechanisms are intricately interconnected and reliant on developmental pathways, including mTOR signaling, Wnt/β-catenin signaling, and energy metabolism [46].

As an essential regulator of tissue homeostasis and regeneration, Wnt/β-catenin signaling is tightly controlled in spatiotemporal patterns [48]. Consequently, any dysregulation of the Wnt/β-catenin pathway is linked to various diseases, including cancer, alopecia, aberrant wound healing, and osteoporosis [49,50,51]. A pivotal aspect of this pathway involves controlling the level of transcriptionally active β-catenin through the APC/Axin/Gsk3β destruction complex, which is essential for the expression of Wnt target genes [13]. This β-catenin destruction complex is composed of Axin, APC, Gsk3β, and casein kinase 1 (CK1). In the absence of Wnt, β-catenin undergoes sequential phosphorylation by CK1 and Gsk3β, leading to ubiquitination and subsequent degradation by the proteasome [52]. Therefore, APC/Axin/Gsk3β destruction complex is one of the important targets for regulating Wnt/β-catenin signaling.

Herein, β-catenin interaction with TCFs/LEFs activated the transcription of target genes such as c-Jun, Lef1, and Myc [37] (Figure 3). In this study, escin-mediated Wnt/β-catenin signaling activation was interfered with by IWR1, but not by DKK1 or Wnt-C59 (Figure 5). DKK1 is a well-characterized LRP5/6 inhibitor that suppresses canonical Wnt signaling by interfering with ligand-receptor interactions at the cell surface [39]. Similarly, Wnt-C59, a PORCN inhibitor, effectively blocks canonical Wnt signaling by preventing the palmitoylation of Wnt3a, a critical modification required for Wnt ligand secretion and activity [40]. However, in this study, the escin-induced activation of Wnt signaling was inhibited only by IWR-1, while neither DKK1 nor Wnt-C59 attenuated this effect. It could be suggested that escin have potential to regulate the stability of the APC/Axin/Gsk3β destruction complex, rather than the Frizzled/Lrp receptor complex (Figure 4 and Figure 6). Furthermore, these findings suggest that escin may activate a non-canonical Wnt/PCP pathway, potentially through JNK activation, as observed in Figure 3c (Figure 6). Further investigation will be required to clarify the precise molecular mechanisms underlying this signaling specificity.

To enhance cutaneous circulation, reinforcement of blood vessels is essential. Vascular remodeling is necessary to create an effective system that can withstand shear stress and deliver oxygen and nutrients to the various parts of the body [12]. In both previous reports and the current study (Figure 5), the activation of Wnt/β-catenin signaling has been indispensable for vessel pruning and network construction [53]. Wnt signaling regulates fundamental aspects of vascular development, including migration, intercellular junctions, and maturation [13].

The JNK pathway has also been reported to contribute to vascular remodeling through the RhoA/ROCK in Wnt/PCP (planar cell polarity) signaling pathway [54] and Integrin-YAP/TAZ signaling cascade [55] (Figure 6). Endothelial cells then develop specialized characteristics to meet the unique requirements of the organs. These include producing and binding chemokines, controlling leukocyte traffic, regulating blood flow, expressing certain transcellular transport systems, controlling permeability, and expressing membrane adhesive molecules [56].

In previous reports, blood vessel remodeling with capillary pruning and/or splitting have been highly correlated to microvasculature and its pathophysiology of microcirculation [53,57]. The in vitro cellular responses related with developmental processes, such as Wnt, BMP, and Notch signaling, were reported to regulate vascular remodeling [58]. However, the in vivo physiological study of the regulating microcirculation, especially in humans, is limited. An elongated examination of the response of increasing blood flow through escin treatment could help to understand the molecular physiology of microcirculation.

Among escin-mediated activation of the pathways, Wnt/β-catenin and JNK signaling (Figure 1 and Figure 2) are especially implicated in cutaneous diseases like androgenetic alopecia, premature hair graying, and edema, all of which are closely related to the quality of life of affected patients [59,60,61,62,63,64]. Furthermore, Wnt/β-Catenin signaling is significantly correlated with the hair follicle cycle, and one of the major therapeutic targets of alopecia treatment, such as androgenic alopecia and alopecia areata [15]. Therefore, future studies about investigating hair follicle physiology with escin and its derivatives would help to find a therapeutic strategy of alopecia treatment.

## 5. Conclusions

In this study, escin was shown to activate Wnt/β-catenin and JNK signaling pathways in HUVECs, with GSK3β involvement confirmed by inhibitor studies using IWR1. Escin enhanced blood vessel remodeling, and its administration prolonged cutaneous blood flow in vivo. Collectively, these findings indicate that escin promotes vascular network formation through GSK3β down-regulation and activation of Wnt/β-catenin and Wnt/PCP signaling pathways, thereby improving microcirculation in the skin. Our results suggest the therapeutic potential of escin for enhancing skin health by improving cutaneous circulation.

## Figures and Tables

**Figure 1 cimb-47-00840-f001:**
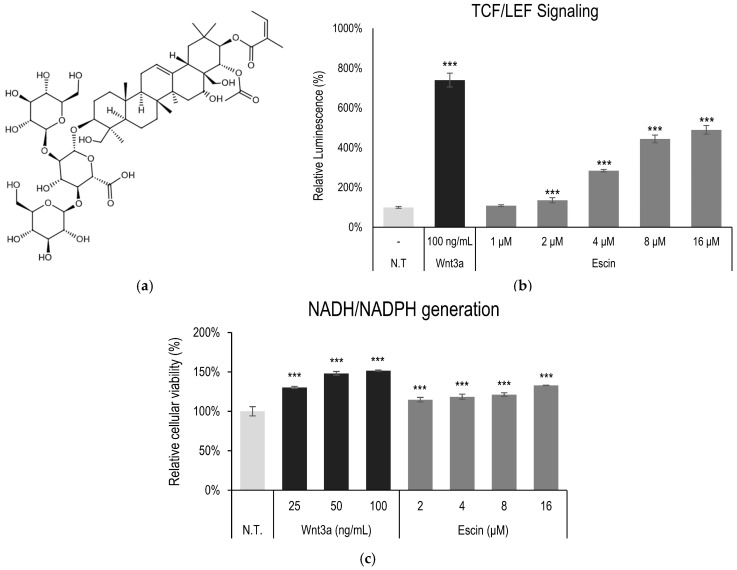
Escin-activated TCF/LEF signaling. (**a**) Chemical structure of escin (**b**) The WRHEK293A cells were treated with escin, and the luciferase activity was measured. (**c**) Cellular viability was examined in HUVEC after escin treatment for 4 h. Wnt3a was used as a positive control. N.T, non-treated control; significantly different compared with N.T (*** *p* < 0.001).

**Figure 2 cimb-47-00840-f002:**
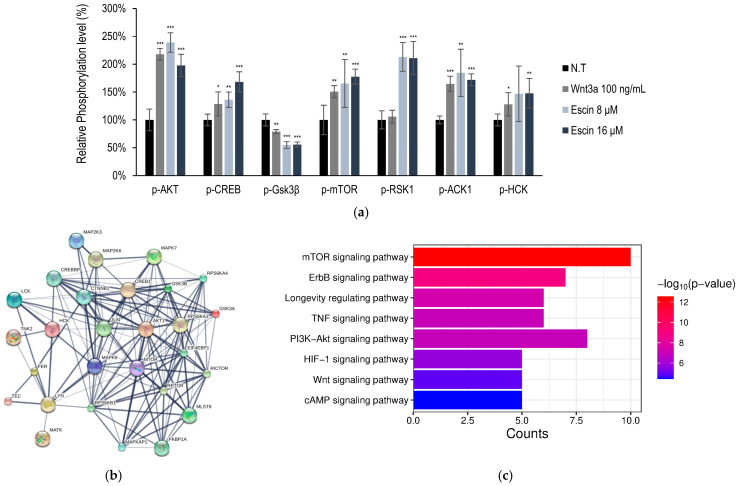
Escin-stimulated Wnt signaling pathway in cultured HUVECs. (**a**) Using anti-phosphor-Tyr antibodies, the relative phosphorylation of seven Tyr kinases was evaluated. (ACK1, AKT, CREB, Gsk3β, mTOR, RSK1, HCK) (**b**,**c**) Protein cluster analysis of genes activated by Escin in HUVEC were investigated. Wnt3a were used as a positive control. N.T, non-treated control; significantly different compared with N.T (* *p* < 0.05, ** *p* < 0.01, *** *p* < 0.001).

**Figure 3 cimb-47-00840-f003:**
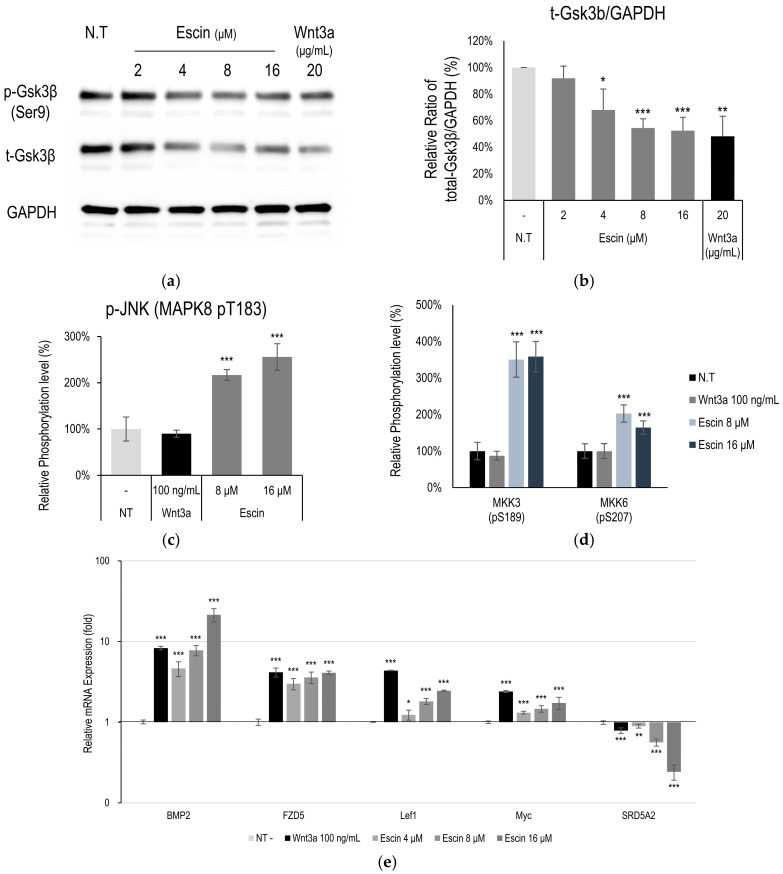
Escin down-regulated Gsk3β in cultured HUVECs. (**a**) The levels of phospho-Gsk3β (Ser 9 site), Gsk3β were further investigated by Western blotting with escin. (**b**) The ratio of Gsk3β/GAPDH was calculated. The data represent the means of five independent samples. Phosphorylation of (**c**) JNK (MAPK8) and (**d**) MKK3, and MKK6 was evaluated. (**e**) As downstream target genes for Wnt signaling, 5 genes (BMP2, FZD5, Lef1, Myc, SRD5A2) were evaluated with escin treatment in cultured HUVECs. The expression level of each gene was measured by RT-PCR. Wnt3a was used as a positive control. N.T, non-treated control; significantly different compared with N.T (* *p* < 0.05, ** *p* < 0.01, *** *p* < 0.001).

**Figure 4 cimb-47-00840-f004:**
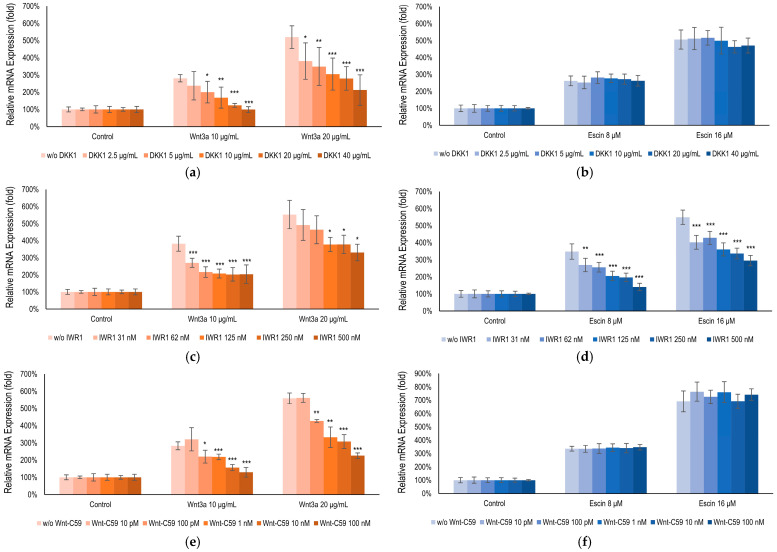
Escin-activated Wnt/β-Catenin signaling via Gsk3β. BMP2 gene expression, a gene activated by both (**b**,**d**,**f**) Wnt3a and (**a**,**c**,**e**) escin, ws examined with co-treatment of Wnt signaling inhibitors; (**a**,**b**) DKK1, (**c**,**d**) IWR1, and (**e**,**f**) Wnt-C59. Control, non-treated control; significantly different compared with each Wnt3a or Escin treated group without inhibitors (* *p* < 0.05, ** *p* < 0.01, *** *p* < 0.001).

**Figure 5 cimb-47-00840-f005:**
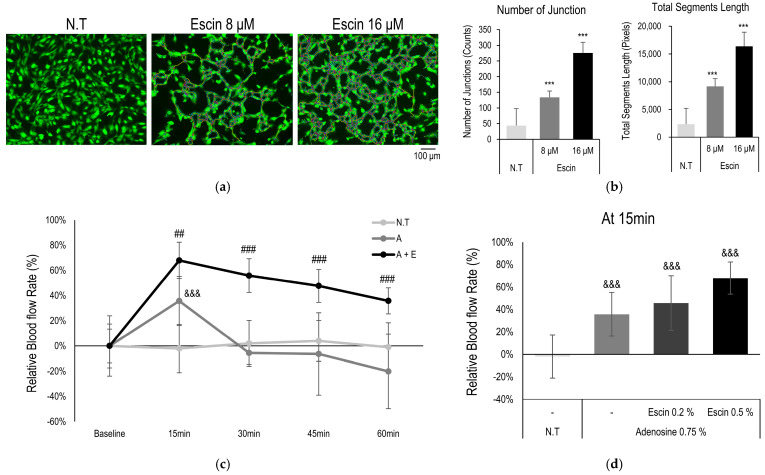
Escin-enhanced blood flow in cutaneous tissue in vivo. Blood vessel remodeling assays were performed with escin treatment. (**a**) Representative images with Calcein staining, and (**b**) image evaluation about number of junctions and total segments length. N.T, non-treated control; significantly different compared with N.T (*** *p* < 0.001). (**c**) Blood flow in human skin was measured with escin (E, 0.5%) and adenosine (A, 0.75%) treatment. (**d**) Relative increase rate of the microcirculation was evaluated at 15 min. significantly different compared with baseline (Start) (&&& *p* < 0.001) and compared with adenosine at each time point (15, 30, 45 and 60 min) (## *p* < 0.01 ### *p* < 0.001).

**Figure 6 cimb-47-00840-f006:**
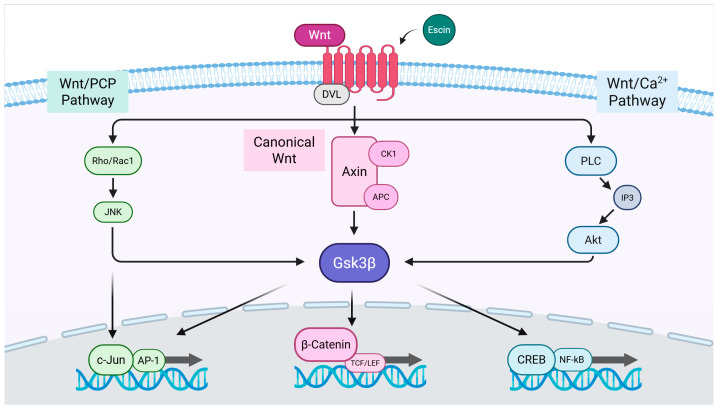
The proposed signal transduction pathways for escin-induced Wnt signaling activation.

**Table 1 cimb-47-00840-t001:** Analysis of correlation of genes in silico with genes activated by escin in vitro (HUVEC).

#node1	#node2	Combined Score	#node1	#node2	Combined Score	#node1	#node2	Combined Score
c-Jun	AKT1	0.977	CTNNB1	AKT1	0.999	RICTOR	AKT1	0.999
	CREBBP	0.999		Gsk3β	0.999		EIF4EBP1	0.999
	CTNNB1	0.992		CREBBP	0.999		RPS6KB1	0.999
	Gsk3β	0.999		Gsk3α	0.988		MAPKAP1	0.999
	MAP2K3	0.78		FER	0.977		RPTOR	0.998
	MAPK7	0.982		RPS6KB1	0.684		Gsk3β	0.929
	MAPK8	0.999		RPS6KA4	0.55		RPS6KA4	0.76
	mTOR	0.775		LCK	0.407		MAPK7	0.436

## Data Availability

The raw data supporting the conclusions of this article will be made available by the corresponding author on request.

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
