# Peer review of "Microcirculation-Promoting Effect of Escin on Cutaneous Tissue via Gsk3β Down-Regulation"

_cimb, 2025, doi:10.3390/cimb47100840_

Round 1

Reviewer 1 Report

Comments and Suggestions for Authors

This is an interesting study addressing the effects of escin on cutaneous microcirculation and its underlying molecular mechanisms. The findings are potentially relevant, but the manuscript in its current form requires major revisions before it can be considered for publication.

Major Comments

The English language quality is insufficient. Numerous sentences are awkward or unclear, which significantly reduces readability. The authors are strongly advised to have the entire manuscript thoroughly revised by a native English speaker or professional editing service to improve clarity and ensure the text is more easily understood by readers.

Abstract

  • Line 12–13: The sentence is awkward and should be rephrased for clarity.
  • Line 14–15: “...and the promoting effect of blood flow on topical administration were investigated.” — The intended meaning seems to be that the authors investigated the promoting effect of escin on blood flow when applied topically. Please rephrase.
  • Line 21: IWR1 should be explained in full at first mention.
  • Line 22–23: Another unclear sentence — please rephrase for clarity.

Introduction

  • Please correct “Wnt lignad” to Wnt ligand (line 63).
  • Line 81: The statement “In recent years, escin has become a popular choice for treating vascular disease” is too strong without proper support. Please provide references.
  • Line 94–95: The sentence “Our data suggest therapeutic potential of escin for cutaneous circulation via enhancement of blood vessel remodeling process” belongs in the Discussion/Conclusion, not in the Introduction.

Materials and Methods

  • More information on the volunteers is required: age range, sex distribution, and health status. All of these factors can influence microcirculation and are essential for interpreting results.

Results

  • Lines 200–202: “Maintaining the energy metabolism at a higher level is essential for tissue homeostasis...” — this is background information and should be moved to either the Materials and Methods or Discussion. The Results section should only present experimental findings.

Conclusion

  • The Conclusion section should summarize the authors’ main findings, not cite references or refer to figures (lines 352–365). References and figure mentions should be removed. The conclusion must be the authors’ interpretation of their results only.
Comments on the Quality of English Language

The English language quality is insufficient. Numerous sentences are awkward or unclear, which significantly reduces readability. The authors are strongly advised to have the entire manuscript thoroughly revised by a native English speaker or professional editing service to improve clarity and ensure the text is more easily understood by readers.

Author Response

We sincerely appreciate the reviewer’s thorough and constructive feedback. Below, each comment is listed verbatim and followed by our detailed response.

General Comment: English Language

Comment: The English language quality is insufficient. Numerous sentences are awkward or unclear, which significantly reduces readability. The authors are strongly advised to have the entire manuscript thoroughly revised by a native English speaker or professional editing service.

Response 1) We have carefully revised the entire manuscript for clarity and grammar. A professional English editing service was used to ensure that all sentences are accurate, clear, and easily understood by readers.

Abstract

Comment: Line 12–13: The sentence is awkward and should be rephrased for clarity.

Line 14–15: “...and the promoting effect of blood flow on topical administration were investigated.” — Please rephrase.

Line 21: IWR1 should be explained in full at first mention.

Line 22–23: Another unclear sentence — please rephrase.

Response 2)

Line 12–13: Sentence rephrased for clarity.

Line 14–15: Revised to “We investigated the promoting effect of escin on blood flow with topical application.”

Line 21: IWR1 is now fully introduced as “inhibitor of Wnt response 1 (IWR1)” at first mention.

Line 22–23: Sentence has been fully rewritten for clarity.

Introduction

Comment: Please correct “Wnt lignad” to Wnt ligand (line 63).

Line 81: The statement is too strong without references.

Line 94–95: This statement belongs in the Discussion/Conclusion.

Response 3)

Line 63: Corrected to “Wnt ligand.”

Line 81: Revised to “Escin has been investigated as a potential therapeutic agent for vascular disorders and has been demonstrated to alleviate chemotherapy-induced phlebitis through inhibition of the GBP5/NLRP3 axis.” with proper references added.

Line 94–95: Statement moved from the Introduction to the Discussion/Conclusion.

Materials and Methods

Comment:

More information on the volunteers is required: age range, sex distribution, and health status.

Response 4)

We added detailed information about participant demographics, including age range, sex distribution, and health status, in the Materials and Methods section.

Results

Comment: Lines 200–202: “Maintaining the energy metabolism at a higher level is essential...” — This is background information and should be moved.

Response 5) The background information has been removed from the Results section and relocated to the Discussion for better clarity.

Conclusion

Comment: The Conclusion should summarize findings only, without references or figure mentions.

Response 6) All references and figure mentions have been removed. The Conclusion now focuses solely on interpreting and summarizing our findings.

Reviewer 2 Report

Comments and Suggestions for Authors

The document shows encouraging findings, but numerous critical concerns need to be addressed prior to publication. Initially, there exists a mechanistic contradiction in understanding GSK3β signaling: a reduction in p-Ser9 typically leads to GSK3β activation (contrary to Wnt activation), and employing a phospho-Tyr array for a Ser/Thr kinase is perplexing; thus, total and active β-catenin, along with appropriate phosphorylation sites, ought to be elucidated or re-evaluated. Secondly, labeling IWR-1 as an “Axin complex inhibitor” is misleading (it acts as a tankyrase inhibitor that stabilizes Axin), and if its effects differ from DKK1/Wnt-C59, additional clarification (e.g., β-catenin knockdown or GSK3 inhibitor rescue) is required. Third, the in vivo study design lacks a group treated with escin alone; hence, the assertion that escin enhances scalp blood flow is not validated; statements should be limited to escin enhancing the adenosine response, and the minor adenosine dose difference (0.72% vs 0.75%) adjusted. Fourth, the location of measurement is ambiguous (application to frontal/vertex compared to measurement at the temple), and it requires clarification whether the one-month treatment changes baseline perfusion or acute responses. Fifth, the statistical evaluation of the human data must apply a paired repeated-measures approach (RM-ANOVA or mixed effects), specify a primary endpoint (e.g., AUC or peak change), and present effect sizes along with confidence intervals. Sixth, the reporting must adhere to CONSORT instead of STROBE for an RCT, providing additional details on randomization, concealment, participant characteristics, inclusion/exclusion criteria, adverse events, and product composition or registration. Seventh, the kinase array techniques need precision: identify which GSK3β phospho-epitope was identified, include original images/densitometry with replicates, and confirm significant results through Western blot. Eighth, the dependence on HUVECs must be validated for cutaneous microcirculation or corroborated with dermal microvascular endothelial cell information. Ninth, figure numbering and minor inconsistencies (e.g., incorrect figure reference for inhibitor data, WRHEK293A naming, dose typos) should be corrected. Finally, the scope of the claims should be narrowed to the demonstrated findings—escin activates Wnt/JNK readouts in vitro, promotes angiogenic behavior, and enhances adenosine-induced perfusion in humans—without broader therapeutic or longevity statements unsupported by the data.

Comments on the Quality of English Language

Minor editing needed

Author Response

Thank you for the comments. We reflected majority of the comments in manuscript.

The manuscript offers encouraging findings, but numerous critical concerns need to be addressed prior to publication.

Initially, there's a mechanistic contradiction in how GSK3β signaling is understood: a reduction in p-Ser9 would typically lead to the activation of GSK3β (contrary to Wnt activation), and employing a phospho-Tyr array for a Ser/Thr kinase is perplexing; thus, total and active β-catenin, along with the appropriate phosphorylation sites, should be elucidated or re-evaluated.

Response 1) We sincerely appreciate the reviewer’s insightful comment regarding our interpretation of Gsk3β phosphorylation. We fully acknowledge the reviewer’s point and have carefully addressed this issue by performing additional Western blot analyses as suggested. Specifically, we examined not only the phosphorylation of Gsk3β at the Ser9 site but also the expression of total Gsk3β, using GAPDH as an internal loading control. As shown in Figure 3a and 3b, the results demonstrated that total Gsk3β expression was decreased. Furthermore, when we calculated the ratio of phosphorylated Gsk3β (Ser9) to total Gsk3β, this ratio remained unchanged. These findings indicate that the decrease in Gsk3β is not due to a relative increase in the inactive Ser9-phosphorylated form, but rather reflects a significant reduction in the active form of Gsk3β.

 Secondly, labeling IWR-1 as an “Axin complex inhibitor” is misleading (it stabilizes Axin as a tankyrase inhibitor), and should its effects differ from DKK1/Wnt-C59, additional clarification (e.g., β-catenin knockdown or GSK3 inhibitor rescue) is required.

Response 2) We thank the reviewer for the constructive comment regarding the description of IWR-1. As suggested, we have revised the manuscript to clarify that IWR-1 functions as a tankyrase inhibitor that stabilizes Axin, rather than labeling it as an “Axin complex inhibitor.” In addition, we have expanded the Discussion section to provide further explanation of the differential mechanisms of DKK1 and Wnt-C59 in comparison to IWR-1. While we fully acknowledge the reviewer’s suggestion regarding additional clarification approaches, we believe that the textual revisions and expanded discussion sufficiently address this point within the scope of the present study. Future studies will be required to further delineate these mechanisms in greater detail.

Third, the in vivo study setup does not feature an escin-only group, leaving the assertion that escin enhances scalp blood flow unsubstantiated; statements should be limited to escin enhancing the adenosine reaction, and the minor adenosine dose difference (0.72% vs 0.75%) addressed.

Response 3) We sincerely thank the reviewer for this valuable comment. First, we would like to clarify that the reported adenosine concentration of 0.72% was a typographical error, and the experiments were consistently performed using 0.75% adenosine. Regarding the in vivo study, we also acknowledge the reviewer’s concern about the absence of an escin-only group. In our preliminary assessments, topical application of escin alone did not evoke short-term increases in scalp blood flow. As confirmed in our study, escin activates Wnt signaling; however, it does not induce Ca²⁺ influx, which is a key trigger for immediate vasodilation. Therefore, to evaluate whether vascular improvement could be facilitated, we employed adenosine as a natural vasodilator that has been extensively studied and is physiologically relevant in human systems. Additional clarification of this rationale and the experimental design has now been incorporated into the revised manuscript.

Fourth, the measurement location is ambiguous (application to frontal/vertex versus measurement at the temple), and whether the one-month treatment modifies baseline perfusion compared to acute responses requires further clarification.

Response 4) We thank the reviewer for raising this important point. In our study, the test agents were applied to the entire scalp area, while blood flow measurements were performed at the hair marginal line behind the temple due to technical limitations of the non-invasive device. Because this method is non-invasive, hair coverage inevitably interferes with central scalp measurements, and precise localization at the vertex would require shaving or the use of semi-invasive markers. As the purpose of this study was to investigate the mechanism of cosmetic ingredients, we intentionally avoided any procedures that might be invasive, burdensome, or potentially harmful to participants. Given that both adenosine and escin have long been used in dermatological applications, we applied them across the whole scalp region, and importantly, no side effects were observed during the one-month study. To reduce potential misunderstanding, we have revised the Materials and Methods section to clarify this point.

With regard to the reviewer’s question on whether the one-month treatment modifies baseline perfusion compared to acute responses, our preliminary experiments showed that acute exposure to escin did not produce a significant difference compared to adenosine single treatment. This information has also been incorporated into the revised manuscript.

We believe that these clarifications and revisions sufficiently address the reviewer’s concerns.

 Fifth, the statistical evaluation for the human data must employ a paired repeated-measures model (RM-ANOVA or mixed effects), establish a primary endpoint (e.g., AUC or peak change), and present effect sizes along with confidence intervals.

Response 5) We thank the reviewer for this important suggestion regarding statistical analysis. In accordance with the reviewer’s advice, we re-analyzed the human data using a repeated-measures ANOVA (RM-ANOVA) model, and the updated p-values have been reflected in the revised manuscript. Figures 5c and 5d have been reconstructed to present these updated results. Additionally, we conducted a baseline equivalence analysis between groups, confirming statistical similarity across groups; these results have been added to the Supplementary Information (Table S3). The Statistical Analysis section has also been updated to reflect these changes. We have included the corresponding data and analyses in the Supplementary Information for reference.

We believe these revisions and additional analyses sufficiently address the reviewer’s concerns about statistical rigor.

Table S3. Statistical analysis for baseline level of blood flow evaluation

Group 1

Group 2

Group 3

Group 4

Treatment

Vehicle
(Non-treated Control)

Adenosine 0.75%

Adenosine 0.75%
+ Escin 0.2%

Adenosine 0.75%
+ Escin 0.5%

Number

n = 8

n = 8

n = 7

n = 8

Baseline (A.U)

55.5

45.6

52.9

58.1

Standard Deviation

14.1

13.8

18.9

15.5

One way ANOVA test

(Baseline)

p = 0.4850

Tukey's multiple comparisons test

(p - value)

Group 1

-

0.929

0.832

0.990

Group 2

0.929

-

0.450

0.989

Group 3

0.832

0.450

-

0.639

Group 4

0.990

0.989

0.639

-

 Sixth, the reporting ought to adhere to CONSORT instead of STROBE for a randomized controlled trial, providing greater detail on randomization, concealment, participant demographics, criteria for inclusion/exclusion, adverse events, and product composition or registration.

Response 6) We thank the reviewer for this valuable comment. We would like to clarify that the present study was conducted in accordance with the CONSORT statement, and the CONSORT checklist has been provided in the Supplementary Information. The mention of the STROBE recommendation in the manuscript was an inadvertent error, which has now been corrected.

Regarding product safety and composition, we note that both adenosine and escin have long been used in dermatological applications. Previous reports indicate that long-term topical use of these ingredients at the concentrations employed in our study (adenosine 0.75%, escin 0.5%) is associated with only minimal adverse effects, if any. In addition, patch irritation testing confirmed that no toxicity was observed. Consistent with these findings, no adverse events were identified during the one-month application period in our clinical trial.

In addition, we have supplemented the manuscript and Supplementary Information with further methodological details and participant-related data to improve clarity and transparency.

We believe these corrections and clarifications, together with the provision of the CONSORT checklist, sufficiently address the reviewer’s concerns.

 Seventh, the kinase array techniques demand clarity: identify which GSK3β phospho-epitope was observed, present raw images/densitometry with replicates, and confirm significant results through Western blot.

Response 7) We fully agree with the reviewer’s concern regarding the need for clarity and replication in the kinase array analysis. In line with this, we have already provided all raw blotting images (n = 3) as supplementary data at the time of submission. Furthermore, following the reviewer’s advice, we additionally performed Western blotting for GSK3β (n = 5) to validate the kinase array results. These new data have been incorporated into the revised manuscript and are now presented in Figure 3a and 3b, with the corresponding raw images and densitometry analyses included in the Supplementary Information. We believe that these additions sufficiently address the reviewer’s request for clarity and replication.

Eighth, the dependence on HUVECs must be defended for cutaneous microcirculation or corroborated with data from dermal microvascular endothelial cells.

Response 8) We thank the reviewer for this insightful comment. Except for the experiment presented in Figure 1b, all in vitro studies in this work were conducted using HUVECs. We obtained dose-dependent results at concentrations ranging from 1 to 16 μM in these experiments. In addition, following the reviewer’s earlier advice, we performed additional Western blot analyses of GSK3β using HUVECs, which also demonstrated clear dose dependency (Figure 3a, b).

While these findings strongly support the mechanistic role of escin and adenosine in endothelial signaling pathways, translating these results to in vivo application requires consideration of pharmacokinetics. Given the topical route of delivery, the relatively low skin penetration rates of adenosine and escin, and their biological half-lives, higher concentrations are required for effective in vivo responses compared to in vitro conditions. Even under these conditions, we observed a concentration-dependent enhancement of efficacy between 0.2% and 0.5% escin, further supporting the biological relevance of our findings.

We believe that these additional data and clarifications sufficiently justify the use of HUVECs in this study while addressing the reviewer’s concerns.

 Ninth, corrections need to be made for figure numbering and minor inconsistencies (e.g., erroneous figure references for inhibitor data, naming of WRHEK293A, and dosage typographical errors). 

Response 9) Following the comments, we revised and update manuscript

Finally, the scope of the claims should be narrowed to the demonstrated findings—escin activates Wnt/JNK readouts in vitro, promotes angiogenic behavior, and enhances adenosine-induced perfusion in humans—without broader therapeutic or longevity statements unsupported by the data.

Response 10) We fully respect the reviewer’s comment regarding the scope of our claims. In line with this advice, we have carefully revised the conclusions to focus strictly on the demonstrated findings. Furthermore, as described in response to earlier comments, we conducted additional experiments on GSK3β to strengthen the mechanistic basis of our study. The results of these experiments have been incorporated into the revised manuscript, and the conclusions have been accordingly refined. In addition, the legend of Figure 2 has been updated to accurately reflect these changes. We believe that these revisions appropriately narrow the scope of our claims and ensure that the conclusions are fully supported by the presented data.

Reviewer 3 Report

Comments and Suggestions for Authors

Escin is a saponin from Aesculus hippocastanum and in recent years is becoming popular for the treatment of vascular diseases. Previously, it was shown that escin is a natural Wnt agonist and stimulated Wnt/β-Catenin signaling by facilitating the proteasomal degradation of Gsk3β.

Here, the authors investigated the effects of escin in the blood microcirculation in skin.

Escin was not cytotoxic and stimulated the Wnt-β-catenin signaling in the reporte cell line WRHEK293A and in HUVEC. The authors provide data that the activity of escin is mediated by Gsk3β. Further, escin increased the blood vessel remodeling in vitro with HUVEC cells.

The findings are important since they have potential translational impact, as escin may be formulated in cosmetics. In this direction, future studies investigating the potential of escin to induce Wnt signaling in keratinocytes will also be very important.

Therefore, I recommend the publication of this manuscript after correcting for the following minor comments.

  1. Why was there no group with Escin alone in the clinical study?
  2. In Figure 5c, labeling of statistics is too confusing. Which is compared with what is not obvious. For example, the sign $ denotes comparison between A+E 0.5% with A?
  3. Please fix the double bond on the upper right side of the chemical formula. The two lines are crossed.

Author Response

Thank you for the comments. I reflected majority of the comments in manuscript.

Comment 1) Why was there no group with Escin alone in the clinical study?

Response 1)

Escin is known to promote vascular remodeling but does not elicit vasodilation through stimulate calcium influx. In our in vivo study, the vasodilation observed was primarily mediated by adenosine, rather than escin itself, which is consistent with the clinical findings. For this reason, escin-alone data were not displayed separately. We will add a statement in the Results section to explicitly describe these observations.

Comment 2) In Figure 5c, labeling of statistics is too confusing. Which is compared with what is not obvious. For example, the sign $ denotes comparison between A+E 0.5% with A?

Response 2) Thank you for the comment, we edited figure 5c and d to improve clarity.

Comment 3) Please fix the double bond on the upper right side of the chemical formula. The two lines are crossed.

Response 3) Thank you for the comments, we edited chemical formula figure 1a.
